# Effects of Health Status, Depression, Gerotranscendence, Self-Efficacy, and Social Support on Healthy Aging in the Older Adults with Chronic Diseases

**DOI:** 10.3390/ijerph19137930

**Published:** 2022-06-28

**Authors:** Hee-Kyung Kim, Jeong-Hyo Seo

**Affiliations:** 1Department of Nursing, Kongju National University, Gongju 32588, Korea; hkkim@kongju.ac.kr; 2Department of Nursing, Graduate School, Kongju National University, Gongju 32588, Korea

**Keywords:** healthy aging, elderly, health status, depression, gerotranscendence, self-efficacy, social support

## Abstract

Purpose of this study is to analyze factors affecting the healthy aging of the elderly with chronic diseases living in the community according to the worldwide aging phenomenon in line with the WHO’s healthy aging strategy. The subjects were 116 elderly aged 65 years or up with one or more chronic diseases and residing in four cities. The collected data were analyzed by using the descriptive statistics, *t*-test, ANOVA, Pearson’s correlational coefficients, and stepwise multiple regression. The healthy aging of subjects showed positive correlations with the health status (r = 0.68, *p* < 0.001), gerotranscendence (r = 0.64, *p* < 0.001), self-efficacy (r = 0.65, *p* < 0.001), and social support (r = 0.47, *p* < 0.001), while the healthy aging and depression (r = −0.58, *p* < 0.001) showed a negative correlation. The factors affecting the healthy aging were health status (*β* = 0.24, *p* = 0.004), self-efficacy (*β* = 0.28, *p* < 0.001), education (*β* = −0.11, *p* = 0.057), exercise (*β* = 0.17, *p* = 0.003), gerotranscendence (*β* = 0.22, *p* = 0.004), and depression (*β* = −0.19, *p* = 0.009), and the explanatory power of those variables was 68.2%. Thus, it would be necessary to provide an intervention for the elderly that could habituate health-related education and exercise, maintain good health status, lower depression, aid control of themselves through the gerotranscendence, and increase self-efficacy.

## 1. Introduction

The aging population is rapidly increasing in the whole world and, especially in Korea, the aged population has remarkably increased, so the country has entered an aged society much more quickly than predicted by the government. As a major social problem that is rapidly progressing, aging means the change of population structure in which the percentage of the aged population is increasing, and the percentage of the youth population is decreasing within the whole population. To prepare for this aging, the international society, including the WHO, selected the plan for life-course approach to healthy aging through the World Health Assembly in 2016, and each country is fulfilling various health-promotion activities for healthy aging of older adults [1]. 

Due to the rapid aging in Korea, it is forecasted that older adults aged 65 years or up would be 15.7% in 2020 and 43.9% in 2060. In 2025, it is projected to be 20.3%, so Korea would enter a super-aged society. Especially, compared to the whole population, the medical expenses per person and out-of-pocket medical expenses of the elderly would be 2.9 times and 2.8 times higher, respectively, so there should be systematic healthcare and welfare policies for older adults [2]. Further, more than 40% of the elderly evaluate their own health as moderate or bad; the elderly who comprise single-person households are over 50% (55.0%); and more and more older adults regard their health status as bad as they get older [3], so healthcare providers such as nurses of the primary healthcare institutions such as health centers in community should urgently provide education and management for healthy aging of the elderly regarding the physical, cognitive/mental, and social aspects. 

In the current era of centenarians, the healthy old age of older adults has become an important goal of life. Thus far, the concept that refers to the healthy old age of older adults has been mixed with the terms “active aging” and “successful aging”, and there have been discussions about the methods and strategies for older adults to live the rest of their lives well. As healthy aging was conceptualized after 2016, the whole world is making various efforts toward this end [1]. As the process of developing and maintaining the functional abilities that enable the elderly to keep their health, healthy aging helps slow down aging physically, psychologically, and socially [4] by maintaining physical/psychological health and continuing social activities [5]. Thus, the elderly’s attitude toward aging could have direct and indirect effects on their health behavior or quality of life, which could be a key point for healthy and successful old age [6]. When the elderly pursue healthy aging based on the attitude to remain healthy in old age, they are valuable, have abilities to do something valuable, make decisions by satisfying basic needs, and have an ability to contribute to society [1]. 

Thus, as the direction of life that elderly should pursue, more research is required regarded healthy aging and its practical application. First of all, in order to improve the healthy aging of older adults in Korea, it would be needed to understand its degree and then to analyze the variables related to it. However, the research on the health of older adults in the nursing aspect has been limited to the concept of attitude towards aging [7], life satisfaction [8], quality of life, health-related quality of life, successful aging, and health conservation [9,10,11,12], while there are not many studies on healthy aging [13]. Especially, a study on healthy aging was conducted a long time ago when the environment was quite different from the recent environment [13], and it did not clearly divide older adults, so it needs to be analyzed more systematically. 

Therefore, for the pursuit of health in the physical, psychological, and social areas of older adults, the relevant factors were reviewed based on the concept of healthy aging as defined by the WHO. 

First, as a factor related to healthy aging in the physical aspect of older adults with chronic diseases living in the community, health status was included. According to the results [14] of a research reporting that it would be more proper and desirable to judge/measure the health status of the elderly, including physical functions, as perceived subjectively in daily life rather than simply calculating the number of diseases under the medical judgment compared to other age groups, health status was included as a variable. 

Next, considering healthy aging in the cognitive/mental aspects of the elderly, depression, gerotranscendence, and self-efficacy were included as variables. When the degree of depression of the elderly was higher, the quality of life was lowered, and cognitive impairment could be caused in older adults with high depression [15]. If the degree of depression was high, it was difficult to fully conserve the health of the elderly [10,12]. As depression showed a negative correlation with the quality of life of older adults with chronic diseases living in the community [9], depression might play a negative role in the maintenance of healthy old age of older adults, so depression was included, as there should be efforts to reduce it. Gerotranscendence showed a positive correlation with the successful aging of older adults, so when the transcendence was higher, aging was performed successfully [11]. Thus, older adults who experienced gerotranscendence perceived aging as a normal human growth/development process and naturally accept the changes of aging [16], so they could experience such healthy aging. The self-efficacy of the elderly showed a positive correlation with quality of life as a main factor affecting quality of life [9]. In the results of a conducted meta-analysis on the variables related to successful aging of Korean people, self-efficacy was included as a psychological variable [17] and had great effects on successful aging of the elderly, so raising self-efficacy would be a very important variable in achieving healthy aging. 

Moreover, as a factor related to healthy aging in the social aspect, social support was included as a variable. Social support was a main factor affecting the successful aging of the elderly in rural areas [18] and also a main factor affecting the quality of life of the elderly with chronic diseases living in the community [9]. Thus, when the elderly more highly perceive the degree of social support they give and receive in their social network with family, relatives, friends, and neighbors, they can obtain more healthy old age. Therefore, social support is regarded as playing a major role in the health and life of older adults [10]. 

Thus, the researcher selected variables such as health status, depression, gerotranscendence, self-efficacy, and social support in the physical, psychological, and social areas that were revealed to be related to healthy aging by organizing the concept of successful aging, quality of life, and health conservation, which were mixed and used to evaluate the health and life of the elderly through the concept of healthy aging that was defined by the WHO, and aimed to analyze their correlations and influences in the integrated level. The purpose of this study is to provide data for the development of intervention programs for improving the quality of life and healthy old age by analyzing the effects of health status, depression, gerotranscendence, self-efficacy, and social support on healthy aging of older adults with chronic diseases. 

## 2. Materials and Methods

### 2.1. Participants

The research subjects totaled 116 older adults aged 65 years or up and with one or more chronic diseases who visited a senior center, a senior college, two senior home-visit centers, a social gathering, and two religious facilities in Daejeon, Gongju, Gangneung, and Sejong cities. Following the criteria of selection, they were male and female elderly people aged 65 years or up who agreed in writing after understanding the purpose of this study and voluntarily showing their intention to participate in this study; had one or more chronic diseases while living in the community; and had abilities to communicate and respond to the questionnaire despite some loss of cognitive functions such as memory. 

The elderly diagnosed with moderate or severe dementia, who could not have dialogues because of no communication ability, could not understand the contents of questionnaire, and who were hospitalized in healthcare facilities such as a long-term care center or hospital were excluded from the subjects. 

In the results of analyzing the number of subjects required in this study by using the G-power 3.1.9.7. Program, a total of 109 samples was needed to maintain a total of eight predictor variables, 0.15 effect size, 0.05 significance level, and 0.80 test power. For the calculation of the number of samples in this study, the method of calculating the number of samples in previous studies for regression analysis was used [11,18]. The eight variables were health status, depression, old age, self-efficacy, social support, exercise, economic status, and education level. Considering the 10% drop-out rate, the data were collected from total 119 people. Excluding three questionnaires with omitted responses, in total, 116 questionnaires were used for the analysis. 

### 2.2. Procedures

The data of this study were collected from 28 April to 22 May 2022. After a meeting, the researchers reviewed a place where many older adults gathered to meet the criteria of the study subjects. The researchers directly visited a senior center, a senior college, two senior home-visit centers, a social gathering, and two religious facilities of Daejeon, Gongju, Gangneung, and Sejong cities; explained the purpose and contents of this study to the president, dean, director of center, or religious leader; and got permission to meet with the subjects for data collection. It took about half an hour. In case the researchers explained the purpose and method of this study to the older adults aged 65 years or up with chronic diseases, and they then agreed to fill out the questionnaire, they were told to fill it out for themselves. Most of the older adults in each place had no difficulty reading or understanding words because they had literacy skills. Researchers encouraged the older adults to read each line of the questionnaire and write down their feelings and thoughts at that time. In the case of the older adults who found it difficult to fill the questionnaire out for themselves, the researchers read the questionnaire to them and helped them respond to it. Before the elderly filled out the questionnaire, they were informed about the purpose and method of this study, confidentiality of personal information, that it would not be used for other than the purpose, and that they could freely stop participating at any time in the middle of survey. It took about 20 min to complete the questionnaire. A prepared souvenir was provided to each subject. 

### 2.3. Measures

#### 2.3.1. Health Status

This study used the tool developed by Speake, Cowart, and Peller [19] and then used by Kim [20]. It was composed of total three items on the basis of a 5-point scale from “Very bad” (1 point) to “Very good” (5 points). The higher score means the higher perception of health status. When it was initially developed, the reliability Cronbach’s α was 0.85, while it was 0.92 in this study. 

#### 2.3.2. Depression

This study used the Geriatric Depression Scale Short Form—Korean that modified the Geriatric Depression Scale by Yesavage et al. [21]. This tool was standardized as suitable for the elderly of Korea by Kee [22], whose version was composed of 10 positive sentences and 5 negative sentences to be responded to in the form of “Yes” or “No”. In the score of each sentence, the response “Yes” was 0 points, while the response “No” was 1 point. The higher score means a more severe degree of depression. When the tool was initially developed, the reliability of Cronbach’s α was 0.88, while it was 0.73 in this study. 

#### 2.3.3. Gerotranscendence

This study used a total of nine items related to transcendence, a sub-area of the Successful Aging Inventory—Korean, which was designed by Troutman et al. [23] and then adapted into Korean and verified in its reliability and validity by Kim [24]. It was composed of total 10 items. Based on the 5-point Likert scale from “Not at all” (1 point) to “Very much likely” (5 points), a high score is interpreted as a high degree of gerotranscendence felt by older adults. In the research by Kim [25], the Cronbach’s α was 0.71, and it was also shown as 0.71 in this study. 

#### 2.3.4. Self-Efficacy

This study used the general self-efficacy scale developed by Chen, Gully, and Eden [25] and then adapted by Noh [26]. It was composed of a total of 8 items. Based on the 5-point Likert scale from “Not at all” (1 point) to “Very much likely” (5 points), a higher score means a higher degree of self-efficacy. In the research by Noh [26], the reliability Cronbach’s α was 0.83, while it was shown as 0.93 in this study. 

#### 2.3.5. Social Support

This study used the self-reported assessment tool called MSPSS (Multidimensional Scale of Perceived Social Support) developed by Zimet et al. [27] and then used as a social support tool by Lee [28]. The tool was composed of 12 items in total about support from family, support from friends, and special support from meaningful others. Based on the 5-point Likert scale from “Not at all” (1 point) to “Very much likely” (5 points), a higher score means higher social support. When the tool was initially developed, the reliability coefficient Cronbach’s α was 0.88, while it was shown as 0.91 in this study. 

#### 2.3.6. Healthy Aging

This study used the healthy aging tool developed as suitable for the elderly of Korea by Ko [29]. It was composed of total 20 items including six items about physical healthy aging, six items about cognitively/mentally healthy aging, and eight items about social support-related healthy aging. Based on the 5-point Likert scale from “Not at all” (1 point) to “Very much likely” (5 points), a higher score means a higher degree of healthy aging. When the tool was initially developed, the correlation coefficient Cronbach’s α was 0.89, while it was shown as 0.92 in this study. The sub-areas were shown as physically healthy aging (0.61), cognitively/mentally healthy aging (0.86), and social support-related healthy aging (0.91). 

### 2.4. Statistical Analyses

The collected data were processed statistically using the SPSS/WIN 25.0 program, and the data analysis method is as follows:Descriptive statistics, such as the average, standard deviation, frequency, and percentage of the subjects’ general characteristics;The subjects’ health status, depression, gerotranscendence, self-efficacy, social support, and healthy aging were analyzed by range, average, and standard deviation;The differences in healthy aging according to the general characteristics of the subjects were obtained by *t*-test and ANOVA, and the post hoc test was obtained by Scheffe test;The correlations of health status, depression, gerotranscendence, self-efficacy, social support, and healthy aging of the subjects were analyzed using Pearson’s correlation coefficients;Stepwise multiple regression was used to identify factors affecting the healthy aging of the subjects.

### 2.5. Ethical Principles

This study obtained the approval from the Institutional Review Board of K University (IRB No. KNU_IRB_2022-1). During the research period, the guidelines of ethical research were obeyed. The consent form included content about anonymity and confidentiality. The subjects were informed that they could stop participating in the study whenever they wanted without any disadvantages even after voluntarily agreeing to participate in this study. They were also informed that the collected data would be handled according to the Personal Information Protection Act, and best efforts would be put forward for the confidentiality of all personal information obtained through this study. They were also told that the collected data would be stored for three years in a locked cabinet that could be only accessed by the researcher, and it would be discarded by using a shredder after e-coding and statistically analyzing for the anonymity of subjects. 

## 3. Results

### 3.1. General Characteristics of Subjects

The total number of older adults with chronic diseases was 116. The range of age was 65–85 years old. The average age was 68.59 ± 4.35 years, and 69% (80 persons) aged 65–69 accounted for the majority. In regards to gender, there were 80 female older adults (69%). As regards spouses, there were 91 people (78.4%) who were living together with their spouses. As for level of education, there were 50 people (43.1%) who graduated from high school. There were 79 people (68.1%) who had a religion. There were 81 people (69.8%) who indicated having one disease. The mean number of diseases was 1.46 ± 0.81. In the results responding to disease-related information, there were 56 people with high blood pressure, 32 people with joint diseases, and 25 people with diabetes. Regarding the number of diseases for which they were taking medicine, there were 77 people (66.4%) with one disease. There were 71 people (61.2%) with a job. There were 79 people (68.1%) who were not exercising regularly. There were 103 people (88.8%) with middle and high economic level (Table 1). 

### 3.2. Degree of Health Status, Depression, Gerotranscendence, Self-Efficacy, Social Support, and Healthy Aging of Older Adults with Chronic Diseases Living in the Community 

The health status of older adults with chronic diseases was 3.27 ± 0.79 of 5 points; the depression was 0.16 ± 0.16 of 0~1 point; the gerotranscendence was 3.30 ± 0.48 of 5 points; the self-efficacy was 3.61 ± 0.69 of 5 points; the social support was 3.71 ± 0.55; and the healthy aging was 3.41 ± 0.52 of 5 points. The sub-areas were shown as physically healthy aging (3.35 ± 0.51), cognitively/mentally healthy aging (3.41± 0.62), and social support-related healthy aging (3.47 ± 0.64) (Table 2).

### 3.3. Comparison of Differences in Healthy Aging According to the General Characteristics of Older Adults with Chronic Diseases 

Healthy aging according to the general characteristics of older adults with chronic diseases is as follows. There were differences in the degree of healthy aging of subjects according to relationship with spouse (t = 2.02, *p* = 0.046), level of education (F = 7.79, *p* = 0.001), religion (t = 2.67, *p* = 0.009), the number of diseases (t = 3.44, *p* =0.001), the number of diseases for which they were taking medicine (t = 2.21, *p* = 0.031), exercise (t = 4.65, *p* < 0.001), and economic level (t = 4.20, *p* < 0.001). 

In other words, older adults who were living together with a spouse showed a higher degree of healthy aging than those who were bereaved, divorced, separated, and unmarried. The older adults who graduated from high school and university or higher showed a higher degree of healthy aging than those who graduated from middle school or lower. The older adults with a religion showed a higher degree of healthy aging than those with no religion. The older adults with a disease showed a higher degree of healthy aging than those with two or more diseases. 

In addition, older adults who were taking medicine in relation to a disease showed a higher degree of healthy aging than those taking medicine in relation to two or more diseases. The older adults who were exercising regularly showed a higher degree of healthy aging than those who were irregularly exercising or not exercising at all. The older adults with middle or high economic level showed a higher degree of healthy aging than those with low economic level (Table 1).

### 3.4. Relationships between Health Status, Depression, Gerotranscendence, Self-Efficacy, Social Support, and Healthy Aging of Older Adults with Chronic Diseases 

The healthy aging showed positive correlations with health status (r = 0.68, *p* < 0.001), gerotranscendence (r = 0.64, *p* < 0.001), self-efficacy (r = 0.65, *p* < 0.001) and social support (r = 0.47, *p* < 0.001) of older adults with chronic diseases, while healthy aging and depression (r = −0.58, *p* < 0.001) showed a negative correlation. In other words, when the health status of subjects was better, when the degree of gerotranscendence was higher, and when the degree of self-efficacy and social support was higher, the degree of healthy aging was high. On the other hand, when the degree of depression was lower, the degree of healthy aging was high. 

The social support of older adults with chronic diseases showed positive correlations with health status (r = 0.33, *p* < 0.001), gerotranscendence (r = 0.56, *p* < 0.001), and self-efficacy (r = 0.52, *p* < 0.001), while it showed a negative correlation with depression (r = −0.30, *p* = 0.001). In other words, when the health status was better, when the degree of gerotranscendence was higher, when the social support was perceived more, and when the degree of depression was lower, the degree of social support was high. 

The gerotranscendence of older adults with chronic diseases showed a positive correlation with health status (r = 0.53, *p* < 0.001), while it showed a negative correlation with depression (r = −0.31, *p* = 0.001). In other words, when the subject perceived their health status as greater, and when the degree of depression was lower, gerotranscendence was high. 

The depression of older adults with chronic diseases showed a negative correlation with health status (r = −0.63, *p* < 0.001). In other words, when the health status of subjects was better, the degree of depression was low (Table 3).

### 3.5. Factors Affecting the Healthy Aging of Older Adults with Chronic Diseases 

To verify the relative influences of the factors that could explain the healthy aging of older adults with chronic diseases, this study added variables, such as relationship with a spouse, level of education, religion, the number of diseases, the number of diseases for which they were taking medicine, exercise, and economic level, that showed significant differences in healthy aging out of the general characteristics and then analyzed them by changing them into dummy variables. Then, the stepwise multiple regression analysis was conducted by inserting the independent variables such as health status, depression, gerotranscendence, self-efficacy, and social support. 

In the results of examining the plot for the homogeneity of variance test, the homoscedasticity was verified, and the value of Durbin–Watson for verifying the independence of residuals was shown as 1.84, which satisfied the hypothesis of independence. In the results of examining the P-P chart for verifying the independence for verifying the normality of error term, a normal distribution was shown. Additionally, in the evaluation of multicollinearity between independent variables, the tolerance limit was 0.43~0.85, which was higher than 0.1. The variance inflation factor (VIF) of each variable was 1.17~2.32, which was not over 10. Thus, it satisfied the basic hypothesis of homoscedasticity and equal variance of residuals. 

The factors affecting the healthy aging of subjects are as follows. In the results of verifying the relative influences of factors affecting the healthy aging of older adults with chronic diseases, the regression model was statistically significant (F = 42.08, *p* < 0.001). The factors affecting the healthy aging were health status (β = 0.24, *p* = 0.004), self-efficacy (β = 0.28, *p* < 0.001), education (β = −0.11, *p* = 0.057), exercise (β = 0.17, *p* = 0.003), gerotranscendence (β = 0.22, *p* = 0.004), and depression (β = −0.19, *p* = 0.009). The explanatory power of those six variables was 68.2%, and the variable with the biggest influence among them was self-efficacy. Through the results above, the health status, self-efficacy, education, exercise, gerotranscendence, and depression were verified as the variables affecting the healthy aging of older adults with chronic diseases (Table 4). 

## 4. Discussion

This study aimed to provide the basic data for the development of healthy aging intervention program by analyzing the factors affecting the healthy aging of the older adults with chronic diseases (Figure 1). In the results of this study, the degree of healthy aging of subjects was 3.41 points of 5.00; physically healthy aging was 3.35 points; cognitively/mentally healthy aging was 3.41 points; and social support-related healthy aging was 3.47 points. In the research [13] that measured the healthy aging of older adults in Seoul and the capital area by using the same tool, the score was 3.37 points, which supported the results of this study. Moreover, reviewing the contents of the survey on healthy aging in the research [29] that developed the healthy aging targeting a total of 350 older adults of Korea, the degree of practicing physically healthy aging, cognitively/mentally healthy aging, and social support-related healthy aging was verified. The mean score and the score of each area were similar, while the moderate level of healthy aging was perceived to be practiced. The healthy aging model of the WHO aims to maintain optimized health by preventing older adults from reaching functional limitation and delaying the situation of aging as far as possible for healthy aging [30]. Thus, with the extended average lifespan, it would be necessary to provide senior-friendly healthy aging management measures suitable for older adults, so older adults with chronic diseases can obtain healthy old age by preventing decrepitude and maintaining their health even in a long-term state of having diseases [31]. 

There were differences in the degree of healthy aging of subjects according to the relationship with a spouse, level of education, religion, the number of diseases, the number of diseases for which they were taking medicine, exercise, and economic level. It is difficult to directly compare, as there are not many studies that compare differences in healthy aging according to the general characteristics of older adults. However, this study aimed to compare the focuses of research on the quality of life, health-related quality of life, successful aging, and health conservation previously used as concepts similar to healthy aging. Furthermore, reviewing the sub-areas of successful aging, health-related quality of life, and health conservation, health in the physical, psychological, and social sense is emphasized, which could be compared with healthy aging. In the case of older adults with chronic diseases, the results of this study were supported by differences in quality of life, successful aging, and health conservation according to spouse, number of diseases, economic level, education, occupation, religion, and exercise [9,11,32]. In the situation of receiving support from a spouse, it is possible to form a relationship to depend on in old age, which is often filled with solitude and loneliness, so the quality of life is improved, and health is conserved well [9,32]. Further, when the level of education is higher, a vocational activity could be performed, which leads to an increase in economic level. As hardship is lessened in old age, it is possible to enjoy healthy aging. In addition, older adults with a religion could be healthy as they could obtain the well-being on a cognitive/mental level [9,11,32]. Older adults with fewer diseases who continuously exercise regularly may have more opportunities to maintain physical and psychological health and also to pursue social health than older adults in the opposite case. Thus, it is crucial to establish management measures for healthy aging by considering the sociodemographic characteristics of older adults. 

The healthy aging of older adults with chronic diseases showed positive correlations with health status, gerotranscendence, self-efficacy, and social support, while healthy aging and depression showed a negative correlation. The healthy aging and health status of subjects showed a positive correlation. This study was supported, as the health status of older adults showed a positive correlation with health-related quality of life and successful aging [10,11]. The perception of health status means the index of how an individual perceives his/her own overall health status, and each individual shows such attitudes and behaviors through the process of perceiving health [33], which is closely related to enjoying the process of healthy aging.

Gerotranscendence also showed a positive correlation with healthy aging, and in the results of a study on successful aging, gerotranscendence showed a positive correlation with successful aging [11,17], which supported the results of this study. With gerotranscendence, older adults obtain a new perspective by positively viewing the aging itself as a natural process since their viewpoint on themselves and the world is changed. Thus, they have less material interest, feel less obsessed about themselves, feel a sense of closeness between generations, and feel happy and satisfied, so they perceive old age positively. Therefore, the achievement of gerotranscendence is an essential element for maintaining healthy old age [34], so the increase of gerotranscendence should be supported.

Self-efficacy showed a positive correlation with the healthy aging of older adults. When the degree of self-efficacy was higher, old age could be successfully obtained [18]. This represents a major psychological variable group in the meta-analysis [17]. The self-efficacy of older adults living in the community also showed an important relationship with quality of life [9], which supported the results of this study. Generally, in the case of older adults, their physical and psychological functions could be weakened easily, and their health could be easily negatively changed compared to other age groups, so it is difficult for them to achieve healthy aging. In the case that self-efficacy is high, they can have confidence in their abilities and initiatively lead their lives based on courage and a challenge-seeking spirit [35]. Thus, it could be an important variable for healthy aging of the older adults, which could bring about healthy old age.

Next, social support also showed a positive correlation with healthy aging. Social support for older adults was supported by a study showing the positive correlation between quality of life and successful aging [9,18]. When the affection and communication were smooth by maintaining relationships with their spouse, family, relatives, and neighbors, older adults could enjoy the satisfactory emotional exchange and life in old age. However, older adults with a low frequency of social interactions showed low life satisfaction [8]. Thus, in order to increase the level of social support for healthy aging, it would be necessary to encourage them to enhance social inclusion by strengthening their communication with their spouse and forming smooth relationships with important people and also participating in various social activities, such as living programs and leisure activities, by utilizing the community resources [18]. 

Meanwhile, the depression of older adults showed a negative correlation with healthy aging, and this result was similar to the results of the studies [15,36] targeting older adults living in a community, which reported that depression lowered the quality of life of older adults. Even though depression is a common psychological problem for older adults, if it is neglected without management or treatment, it could cause physical, cognitive, and social malfunction [36]. In particular, when physical pain is caused by chronic diseases, quality of life can be lowered because of depression and sleep disorders, and they cannot have healthy old age [37]. Thus, in addition to relieving physical health problems to reduce depression in older adults and increasing their perception of health status [15,37], they should be empowered and supported within social relationships through social activities [36]. 

The factors affecting healthy aging of older adults with chronic diseases were health status, self-efficacy, education, exercise, gerotranscendence, and depression. Among them, the variable with the greatest effect was self-efficacy. In the results of research targeting older adults [13], health status and depression were shown as the factors affecting healthy aging, which supported the results of this study. As positive attitude and perception of health are important to maintain physical and psychological functions, it is necessary to support them in habituating health-promotion behavior. In other words, if they regularly and properly maintain their lifestyle, such as exercise and regular life by maintaining the functional health status in daily life, they could achieve successful aging and healthy aging. Furthermore, as older adults get older, emotional health status can deteriorate. Successful aging could be achieved by selecting an area to be improved, increasing conservable ability, and complementing the loss with learning, external help, and psychological reward, which could guarantee healthy aging. 

Self-efficacy was the factor with the greatest effect on healthy aging. In research targeting older adults living in the community [12], self-efficacy had complete mediating effects on the relationship between depression and health conservation and showed a strong effect on the conservation of health, which was similar to the results of this study. As something to maintain the balance and state of wellbeing and as a factor of physical, mental, and sociopsychological integration [12], health conservation is very similar to the concept of healthy aging in the physical, cognitive/mental, and social areas, which could be applied to discussions. Self-efficacy refers to self-confidence and belief by which an individual can organize and successfully perform a series of actions required to obtain results in a given situation, which plays an important role in the change or continuance of behaviors. People with high self-efficacy can perform and enhance health-related behaviors based on flexible skills to cope with a situation, and eventually, they could achieve higher quality of life and healthy aging [38]. Thus, older adults should be encouraged to make efforts to build confidence and show abilities in order to increase their self-efficacy. 

Next, exercise and level of education were factors affecting the healthy aging. It was said that exercise in older adults affects healthy aging. Especially, exercise behavior and period of participation in exercise had effects on the sub-area of healthy aging such as physical aging [29]. As health-related fitness was an important factor affecting the quality of life of older adults, the importance of exercise should be emphasized [15]. The subjects of this study showed a high degree of healthy aging in the case of exercising regularly and obtaining a high level of education. Maintaining physical strength through exercise could bring about brain health, prevention of depression and decline in cognitive function, and healthy aging. As people with higher education can easily secure economic stability and social status, they can pursue healthy aging. Healthy older adults take care of their own health and pursue the self-value of a healthy life through social relationships based on group exercise [29], so exercise becomes an important element for maintaining healthy old age. Especially, in the case of older adults with chronic diseases living in the community, it would be necessary to encourage them to regularly exercise by forming a gathering focused t a health care center or lead by village leaders. In addition, exercise for muscular strength in lower limbs could improve the muscular strength of older adults and improve their life satisfaction, daily living activity, and quality of life [29], so it would be desirable to provide some exercise programs using this concept. 

Moreover, because the range of social activities and economic ability are decided by differences in high education and low education, the level of education should be considered for healthy aging [29]. However, older adults cannot increase their level of education. Thus, it is crucial to widen the opportunity of knowledge and practice by establishing a healthy aging educational program led by experts and then continuously providing this healthy aging program to them. 

Next, gerotranscendence was a factor affecting healthy aging. Gerotranscendence has a correlation with successful aging, and older adults should be encouraged to acquire gerotranscendence, which is the main psychological factor for successful old age of older adults [11]. Gerotranscendence builds successful aging by utilizing a macroscopic perspective to consider others beyond selfish interest and material values [17]. The older adults who have reached the gerotranscendence can enjoy their lives in an unflappable and comfortable state and also achieve healthy aging by perceiving aging as the normal human growth/development process and accepting changes in physical, psychological, and social aging [11]. Thus, the measures for achieving gerotranscendence should be sought after. 

Next, depression was a factor affecting healthy aging. Depression had negative effects on the quality of life [9,36], successful aging [17], and health conservation [12], which supported the results of this study. Depression in older adults lowers quality of life and life satisfaction, causes difficulties in conserving health well, and prevents them from having successful old age or achieving healthy aging. As an affective disorder accompanied by anxiety, helplessness, and worthlessness caused by changes in affective mood [13], depression is a health problem commonly shown in older adults. As this could be an obstacle to daily life and social life, depression should be managed for the healthy aging of older adults.

In this study, social support had no effects on healthy aging. In the results of verifying the successful aging structural model of older adults in research [39], social health status, including social support, had no direct/indirect effects on successful aging, which was similar to the results of this study. Since research results show that social support is a major influencing factor on healthy aging [10], continuous research on the impact of social support is needed to support the pursuit of healthy aging for older adults in the future.

This study is significant in the aspect of providing data for the development of interventions by considering and comprehensively revealing various factors in the physical, cognitive/mental, and social areas required for providing healthcare services to older adults with chronic diseases, so they could obtain healthy old age and improve their quality of life by pursuing and practicing healthy aging. However, instead of recruiting older adults throughout the whole nation, this study used the convenience sampling method only in some limited areas, so care should be taken in generalizing the results of this study. It is proposed to be expanded and studied in older adults in more regions nationwide.

## 5. Conclusions

This study aimed to analyze the effects of health status, depression, gerotranscendence, self-efficacy, and social support on the healthy aging of older adults with chronic diseases. In addition, the relationship between variables and differences in healthy aging according to general characteristics was analyzed. Health status, self-efficacy, education, exercise, gerotranscendence, and depression were the significant predictor factors on the healthy aging of research subjects, and the explanatory power of those six variables was 68.2%. In this study, control variables, such as gender, relationship with spouse, religion, the number of diseases, and economic level, and the independent variables, such as social support, did not have effects on the healthy aging of subjects. There should be further research utilizing various variables, including those already studied.

In addition, in facing the era of centenarians, the older adults with chronic diseases would need to habituate exercise and health-related learning, have confidence in their ability, and manage their own health based on the factors affecting healthy aging verified in this study. Moreover, they should try to live a transcendental life by psychologically relieving their depressed mood. The state and local governments should actively provide vitalization measures and administrative/financial support, such as healthcare programs focusing on primary healthcare institutions, the practical application of various programs using senior centers and shelters installed in community, and the formation of self-help groups for the healthy aging of local residents. 

## Figures and Tables

**Figure 1 ijerph-19-07930-f001:**
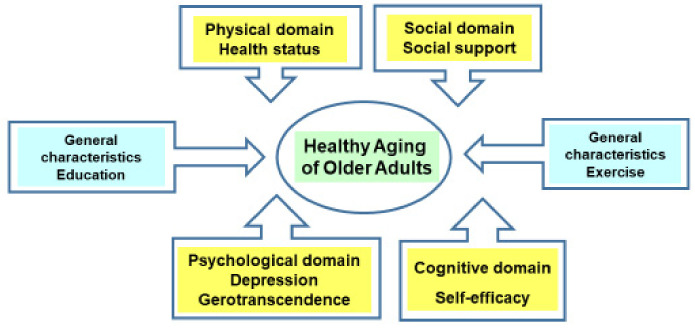
Factors affecting healthy aging.

**Table 1 ijerph-19-07930-t001:** Differences in healthy aging according to general characteristics.

Variables	Classification	*n*	%	Mean	SD	t/F	*p*-Value Scheffe Test
Age (year)	65–69	80	69.0	3.40	0.48	0.22	0.804
	70–79	31	26.7	3.46	0.66		
	≥80	5	4.3	3.31	0.26		
Gender	Male	36	31.0	3.55	0.52	1.82	0.070
	Female	80	69.0	3.36	0.52		
Spouse	Cohabited	91	78.4	3.47	0.54	2.02	0.046
	Bereaved, divorced, separated, unmarried	25	21.6	2.23	0.42		
Education	Graduated from middle school or lower (a)	40	34.5	3.19	0.56	7.79	0.001
	Graduated from high school (b)	50	43.1	3.46	0.42		a < b, c
	Graduated from university or higher (c)	26	22.4	3.68	0.52		
Religion	Yes	79	68.1	3.50	0.53	2.67	0.009
	No	37	31.9	3.23	0.46		
The number of diseases	1	81	69.8	3.52	0.51	3.44	0.001
	≥2	35	30.2	3.17	0.48		
The number of diseases they take medicine for	1	77	66.4	3.50	0.46	2.21	0.031
	≥2	39	33.6	3.25	0.60		
Job	Yes	71	61.2	3.48	0.44	1.50	0.139
	No	45	31.8	3.32	0.63		
Exercise	Regular exercise	37	31.9	3.72	0.56	4.65	<0.001
	No regular exercise	79	68.1	3.27	0.44		
Economic level	Middle and high	103	88.8	3.48	0.50		
	Low	13	11.2	2.88	0.43	−4.20	<0.001

**Table 2 ijerph-19-07930-t002:** Degree of health status, depression, gerotranscendence, self-efficacy, social support, and healthy aging of subjects.

Variables	Mean	SD	Range
Health status	3.27	0.79	1–5
Depression	0.16	0.16	0–0.67
Gerotranscendence	3.30	0.48	2.50–5
Self-Efficacy	3.61	0.69	2–5
Social support	3.71	0.55	2.17–5
Healthy aging	3.41	0.52	2.10–4
Physically healthy aging	3.35	0.51	2.33–4.83
Cognitively/mentally healthy aging	3.41	0.62	2–5
Social support-related healthy aging	3.47	0.64	1.88–5

**Table 3 ijerph-19-07930-t003:** Relationships between health status, depression, gerotranscendence, self-efficacy, social support, and healthy aging of subjects.

Variables	Health Status r (*p*)	Depression r (*p*)	Gerotranscendence r (*p*)	Self- Efficacy r (*p*)	Social Support r (*p*)	Healthy Aging r (*p*)
Health status	1					
Depression	−0.63 (<0.001)	1				
Gerotranscendence	0.53 (<0.001)	−0.31 (0.001)	1			
Self-efficacy	0.42 (<0.001)	−0.36 (<0.001)	0.63 (<0.001)	1		
Social support	0.33 (<0.001)	−0.30 (0.001)	0.56 (<0.001)	0.52 (<0.001)	1	
Healthy aging	0.68 (<0.001)	−0.58 (<0.001)	0.64 (<0.001)	0.65 (<0.001)	0.47 (<0.001)	1

**Table 4 ijerph-19-07930-t004:** Factors affecting the healthy aging of subjects.

Variables	B	SE	β	t	*p*
Constant	1.41	0.24		5.88	<0.001
Health status	0.16	0.05	0.24	3.00	0.004
Self-Efficacy	0.21	0.05	0.28	4.03	<0.001
Education (above graduate high school) *	−0.12	0.06	−0.11	−1.93	0.057
Exercise (doing regularly) *	0.19	0.06	0.17	3.02	0.003
Gerotranscendence	0.24	0.08	0.22	2.92	0.004
Depression	−0.64	0.24	−0.19	−2.66	0.009

SE, standard error; * dummy variable: education (0 = Graduation from middle school or lower, 1 = Graduation from high school and university or higher); exercise (0 = Irregular or no exercise, 1 = Regular exercise).

## Data Availability

The data underlying this article will be shared upon reasonable request from the corresponding author.

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
