# Peer review of "Effects of Health Status, Depression, Gerotranscendence, Self-Efficacy, and Social Support on Healthy Aging in the Older Adults with Chronic Diseases"

_ijerph, 2022, doi:10.3390/ijerph19137930_

Round 1
Reviewer 1 Report
Thank you for the privilege to review your manuscript, which was aimed to investigate factors that affect healthy aging among the elderly with chronic diseases living in the community. The manuscript has potential to be interesting and make a worthwhile contribution to understanding healthy aging. I would like to point and highlight the following:
1. Extensive English language editing would be beneficial to improve readability and presentation of ideas.
2. Line 158... D, G1, G2, and S cities: Is there any compelling reason why the cities cannot be identified? It may be helpful to contextualise the study/findings and elucidate the population and setting. Please explain.
3. Lines 168-172: Please explicitly state or clarify what the primary outcome was. Following this, make the sample size calculations clearer by stating the outcome, with references, which formed the basis for sample size calculations.
4. Lines 264-286: Too long. I would suggest providing a descriptive overview in no more than a quarter of the length of what it is currently. The idea is to cross-reference Table 1 without repeating what is in the table. The same applies to other areas of the manuscript where I would encourage shorter narratives/explanations of key findings rather a complete repetition of what is contained in the tables.
5. Table 1. The table column headings can be improved with better labels which clearly indicate what the reported data are. At the moment the headings are confusing, i.e. reporting "n" and "mean" or % and SD in the same column.
6. Line 293-298 and Table 2: Please explain the choice of reporting these ordinal scales data as means and SD as opposed to Median (Range or IQR).
7. Line 617-619: Exercise and health literacy as referred to in this segment were not evaluated as mediators of healthy aging. As such, the statement is much more suited for the discussion section.
Author Response
Thank you very much for your careful review so that it can be a good paper. As you pointed out, I worked hard to revise it. Thank you so much. .
|
Points to note |
revision |
|
1. Extensive English language editing would be beneficial to improve readability and presentation of ideas. |
I have been reviewed the English description by a professional company again. |
|
2. Line 158... D, G1, G2, and S cities: Is there any compelling reason why the cities cannot be identified? It may be helpful to contextualise the study/findings and elucidate the population and setting. Please explain. |
I wrote the cities full name. |
|
3. Lines 168-172: Please explicitly state or clarify what the primary outcome was. Following this, make the sample size calculations clearer by stating the outcome, with references, which formed the basis for sample size calculations. |
Added content to the 168-172 line.; The eight variables were health status, depression, old age, self-efficacy, social support and exercise, economic status, and education level. |
|
3. Lines 264-286: Too long. I would suggest providing a descriptive overview in no more than a quarter of the length of what it is currently. The idea is to cross-reference Table 1 without repeating what is in the table. The same applies to other areas of the manuscript where I would encourage shorter narratives/explanations of key findings rather a complete repetition of what is contained in the tables. |
Table 1 only describes important content and reduces it a lot. The total number of older adults with chronic diseases was 116. The range of age was 65~85 years old. The average age was 68.59 ± 4.35 years, and 69% (80 persons) aged 65-69 accounted for the majority. In the gender, there were 80 female elderly (69%). In the condition of spouse, there were 91 people (78.4%) who were living together with their spouses. In the level of education, there were 50 people (43.1 %) who graduated from high school. There were 79 people (68.1%) who had a religion. There were 81 people (69.8%) who said one disease. The mean number of diseases was 1.46 ± 0.81. In the results of responding to the diseases, there were 56 people with high-blood pressure, 32 people with joint diseases, 25 people with diabetes. Regarding the number of diseases they were taking medicine in relation to diseases, there were 77 people (66.4%) with one disease. There were 71 people (61.2 %) with a job. There were 79 people (68.1%) who were not exercising regularly. There were 103 people (88.8%) with middle and high economic level (Table 1).
|
|
5. Table 1. The table column headings can be improved with better labels which clearly indicate what the reported data are. At the moment the headings are confusing, i.e. reporting "n" and "mean" or % and SD in the same column. |
In table 1, I deleted the mean and standard deviation included in the number of people and ratio. |
|
4. Line 293-298 and Table 2: Please explain the choice of reporting these ordinal scales data as means and SD as opposed to Median (Range or IQR). |
The tool of this study is not a ranking variable made of one item, but consists of 3 to 20 questions, so you can check the degree of the variable by means, standard deviation, and range. I ask for your understand. Please be generous. |
|
7. Line 617-619: Exercise and health literacy as referred to in this segment were not evaluated as mediators of healthy aging. As such, the statement is much more suited for the discussion section. has potential to be interesting and make a worthwhile contribution to understanding healthy aging. I would like to point and highlight the following: |
I deleted the contents and wrote it anew. In addition, based on the influencing factors identified in this study, it is necessary to increase the confidence that the elderly with chronic diseases can manage their health well by paying attention to their health conditions in the age of 100. Psychologically, you should try to live a superfluous life while relieving your gloomy mood. |

Reviewer 2 Report
This manuscript presents the results of a survey investigating factors related to healthy aging. The results are solid and well presented and will be of great interest to others.
Throughout the manuscript subjects are referred to as elderly or elderly people. The commonly used term is older adults and I recommend it be used instead of" elderly".
The Introduction is too long and has 19 references. Limit the introduction to why you have done this research (including references to results that led you to do your study). But there is no need to discuss in great detail all 19 of those studies in the Introduction. Put that information in the Discussion. This will require that you move the references to the Discussion so they will need to be renumbered.
Your Discussion is too detailed and is redundant with the Introduction. There is no need to discuss in great detail the results of others. Limit your discussion of results of other researchers to whether they agree with your results or not, and why. This is your paper, not theirs. Highlight your results, not that of others.
Author Response
Thank you very much for your careful review so that it can be a good paper. As you pointed out, I worked hard to revise it. Thank you so much. .
|
Points to note |
revision |
|
Throughout the manuscript subjects are referred to as elderly or elderly people. The commonly used term is older adults and I recommend it be used instead of" elderly". |
In the entire sentence, I modified the elderly to older adults. |
|
The Introduction is too long and has 19 references. Limit the introduction to why you have done this research (including references to results that led you to do your study). But there is no need to discuss in great detail all 19 of those studies in the Introduction. Put that information in the Discussion. This will require that you move the references to the Discussion so they will need to be renumbered. |
I've reduced the introduction.
|
|
Your Discussion is too detailed and is redundant with the Introduction. There is no need to discuss in great detail the results of others. Limit your discussion of results of other researchers to whether they agree with your results or not, and why. This is your paper, not theirs. Highlight your results, not that of others. |
Reduce the discussion.
|

Reviewer 3 Report
I would like to congratulate the authors for their interest in researching in this field, however, the work presented presents some deficiencies.
a) The title is appropriate for the article and it is too long and may confuse the reader.
The title should have the following characteristics:
-Describe the content of the article in a specific, clear, accurate, brief, and concise manner.
-Enable the reader to identify the topic easily.
-Allow a precise indexing of the article.
I propose that the authors specify the community in which the subjects under study are located and eliminate "In line with 4 the WHO's Healthy Aging Strategy and Plan”.
Anyway, authors may propose a different modification of the title in accordance with the points mentioned above.
b) Abstract section is correct but the authors have segmented it into background, methods, results,...according to the different chapters of the article. I consider that these references to the different sections of the article should be eliminated
c) Authors are committed to specific purposes in this study (line 142).
These purposes must be corroborated at the end of the study (in the conclusions section) or be revised before the article is ready for submission.
d) The sample used seems to me to be small. Although the authors justify statistically the adequacy of the data collected, I consider that they should indicate this point as one of the limitations of the study.
e) Authors define the participants and procedures in section 2.1 and 2.2. However, I have missed a summary of the survey/interview conducted in order to facilitate the reader's understanding of the completeness of the data collection process.
f) In the results section, I missed the presence of some illustrative figure of the results to help the reader to quickly understand the results obtained. Although this point is not strictly necessary, I consider that it would be interesting to improve the applicability of this research.
I hope that these changes will help to improve your article and make it a document of great scientific interest.
Author Response
Thank you very much for your careful review so that it can be a good paper. As you pointed out, I worked hard to revise it. Thank you so much. .
|
Points to note |
revision |
|
a) The title is appropriate for the article and it is too long and may confuse the reader. The title should have the following characteristics: -Describe the content of the article in a specific, clear, accurate, brief, and concise manner. -Enable the reader to identify the topic easily. -Allow a precise indexing of the article. I propose that the authors specify the community in which the subjects under study are located and eliminate "In line with 4 the WHO's Healthy Aging Strategy and Plan”. Anyway, authors may propose a different modification of the title in accordance with the points mentioned above. |
a) The title has been completely modified and shortened. Effects of Health Status, Depression, Gerotranscendence, Self-efficacy and Social Support on Healthy Aging in the Older Adults with Chronic Diseases
|
|
b) Abstract section is correct but the authors have segmented it into background, methods, results,...according to the different chapters of the article. I consider that these references to the different sections of the article should be eliminated |
I have deleted the background, method, and result from the abstract.
|
|
c) Authors are committed to specific purposes in this study (line 142). These purposes must be corroborated at the end of the study (in the conclusions section) or be revised before the article is ready for submission. |
The specific purpose has been deleted. And I added it to the conclusion. In addition, the relationship between variables and differences in health aging according to general characteristics was analyzed. |
|
d) The sample used seems to me to be small. Although the authors justify statistically the adequacy of the data collected, I consider that they should indicate this point as one of the limitations of the study. |
At the end of the discussion, the limitations of the study, I suggested expanding the elderly population. It is proposed to be expanded and studied for the elderly in more regions nationwide. |
|
e) Authors define the participants and procedures in section 2.1 and 2.2. However, I have missed a summary of the survey/interview conducted in order to facilitate the reader's understanding of the completeness of the data collection process. |
I added the details to the data collection section. Most of the older adults in each place had no difficulty reading or understanding words because they had literacy skills. Researchers encouraged the elderly to read each line of the questionnaire and write down their feelings and thoughts at that time. |
|
f) In the results section, I missed the presence of some illustrative figure of the results to help the reader to quickly understand the results obtained. Although this point is not strictly necessary, I consider that it would be interesting to improve the applicability of this research. I hope that these changes will help to improve your article and make it a document of great scientific interest. |
Added figure 1. for result description. figure1. Effect factors on healthy aging of subjects |

Reviewer 4 Report
Dear authors,
congratulation on your excellent work and for choosing such an important topic. I have few recommendations for your article, as follow:
the title is too long and needs to be shortened. The introduction shows a good overview of what is known about the topic, but it is too long and needs to be shortened. Matter and Methods are well described. It is not necessary to write zero in the Results if it is an integer. i.e. 69.0% should write 69%. The results are well presented, in terms of tables and content. The discussion is written on five pages which is not necessary as it is tedious for the reader. In the discussion, explain only the significant results that should be compared with indicators from other countries.
Author Response
Thank you very much for your careful review so that it can be a good paper. As you pointed out, I worked hard to revise it. Thank you so much. .
|
Points to note |
revision |
|
the title is too long and needs to be shortened. The introduction shows a good overview of what is known about the topic, but it is too long and needs to be shortened. Matter and Methods are well described. It is not necessary to write zero in the Results if it is an integer. i.e. 69.0% should write 69%. The results are well presented, in terms of tables and content. The discussion is written on five pages which is not necessary as it is tedious for the reader. In the discussion, explain only the significant results that should be compared with indicators from other countries. |
The title has been modified briefly and accurately. .0 has been removed in numbers. Thank you. |

Round 2
Reviewer 1 Report
Thank you for your generally satisfactory responses to my original comments and queries. Inasmuch as reviewers we are to focus on the content and technical and not so much on the language aspects of the manuscripts, language has an impact on the ability to provide sound appraisal of quality of manuscript. I do appreciate that you procured the services of a professional English language editing company, however it is still apparent that further work is still required in this area.
Author Response
Thank you for your valuable comment.
I checked it carefully once again.
I modified it. Please take good care of it.

Reviewer 2 Report
DO a global search f of the entire manuscript to replace elderly with older adults. You missed those that are in the Methods and in Results 3.1, 3.2, and 3.3. You also continue to use elderly in the text that you added. Be sure to change them all to older adults.
For Figure 1, I recommend changing the title to "Factors affecting healthy aging", and the circle in the middle of the figure should read "Healthy Aging of older Adults) This table is very well done.
Author Response
I carefully modified the elderly to the older adults from the beginning.
Thank you so much.
